# A New Fluorescent Chemosensor for Cobalt(II) Ions in Living Cells Based on 1,8-Naphthalimide

**DOI:** 10.3390/molecules24173093

**Published:** 2019-08-26

**Authors:** Yu-Long Liu, Liu Yang, Lu Li, You-Quan Guo, Xiao-Xiao Pang, Ping Li, Fei Ye, Ying Fu

**Affiliations:** 1Department of Applied Chemistry, College of Science, Northeast Agricultural University, Harbin 150030, China; 2Heilongjiang Vocational College of Biology Science and Technology, Harbin 150025, China

**Keywords:** 1,8-naphthalimide, colorimetric, fluorescent, cobalt(II) ions, live cell imaging

## Abstract

In this work, a highly selective fluorescent chemosensor *N*-(2-(2-*butyl*-1,3-dioxo-2,3-dihydro-1*H*-benzo[*de*]isoquinolin-6-yl)hydrazine-1-carbonothioyl)benzamide (**L**) was prepared and characterized. An assay to detect the presence of cobalt(II) ions was developed by utilizing turn-on fluorescence enhancement with visual colorimetric response. Upon treatment with Co^2+^, a remarkable fluorescence enhancement located at 450 nm was visible to naked eyes accompanied with a distinct color change (from pink to colorless) in a CH_3_CN/HEPES (4/1, *v*/*v*, pH = 7.4) solution due to the formation of a 1:1 complex at room temperature. In addition, the linear concentration range for Co^2+^ was 0–25 µM with the limit of detection down to 0.26 µM. Thus, a highly sensitive fluorescent method based on chelation-assisted fluorescence enhancement was developed for the trace-level detection of Co^2+^. The sensor was found to be highly selective toward Co^2+^ ions with a large number of coexisting ions. Furthermore, the **L** probe can serve as a fluorescent sensor for Co^2+^ detecting in biological environments, demonstrating its low toxic properties to organisms and good cell permeability in live cell imaging.

## 1. Introduction

Cobalt is a transition metal element present in rocks, soil, plants and animals in trace levels and is widely used in various materials such as alloys [1], batteries [2] and catalysts [3]. As a vital trace element in human body, cobalt plays vital roles in various physiological processes which are essential to the metabolism of all animals [4,5]. Meanwhile, cobalt is a key constituent of cobalamin, also known as vitamin B12, whose insufficiency ultimately gives rise to hematological and neurological disorders [6]. Therefore, cobalt, a trace element in nature, is beneficial to organisms. However, the excessive intake of Co^2+^ by the body can cause toxicological effects including asthma, rhinitis, allergic dermatitis vasodilation and cardiomyopathy in human and animals [7,8]. Furthermore, in view of the toxic effect caused by Co^2+^, it is a significant environmental pollutant, and the maximum acceptable level of Co^2+^ in drinking water is permitted to be 40 µg·L^−1^ (World Health Organization) [9]. Hence, a highly selective and sensitive determination of trace amounts of Co^2+^ is essential and of great importance in biological and environmental samples. 

Several principal analytical methods including inductively coupled plasma atomic emission spectrometry (ICP-AES) [10], flame atomic absorption spectrometry [11], surface plasmon resonance (SPR) [12], surface-enhanced Raman scattering (SERS) [13], chemiluminescence [14] and electrochemical methods [15] have been reported aiming to detect Co^2+^. Unfortunately, these techniques usually require either expensive equipment, cumbersome processes, or tedious pretreatments, making them difficult to apply to the real-time determination of Co^2+^. Due to the limitations of these methods, the development of chemosensors towards Co^2+^ ions with high sensitivity and easy operation is still a significant work for researchers. Colorimetric methods for Co^2+^ detection which can conveniently and easily monitor target ions with the naked eye have attracted considerable attention [16,17,18,19,20,21,22,23,24,25]. Additionally, a fluorescence probe has also evolved to be a promising candidate for detecting Co^2+^ by virtue of its high specificity towards target substances, great simplicity and high-speed analysis during the process of experiments [26,27,28,29]. These chemosensors, however, are based on two independent kinds of response mechanism—colorimetric detection and fluorometric detection. Due to their highly sensitive and selective sensing for target ions with the naked eye, it would be beneficial if the colorimetric and fluorescent methods could be joined together with inexpensive equipment which is convenient and easy to operate. On the other hand, some of these probes suffer from notable toxicity, low bio-compatibility, and complicated synthetic processes, which may hinder their specific application in related fields. Consequently, for the further and efficient detection of Co^2+^, searching for a novel and concise probe is still in urgent demand. 

Fluorophore naphthalimide is widely used for the construction of excellent fluorescent chemosensors because of its excellent photophysical properties such as remarkable chemical stability, large Stokes shift, high fluorescent quantum yields and ease of structural modification [30,31,32,33,34,35,36,37]. Thiourea is generally utilized as a receptor unit for the molecular design of probe for metal ions such as Ag^+^, Hg^2+^, Zn^2+^, Pb^2+^, Fe^3+^ and F^-^ due to its strong chelation with transition metal ions (Scheme 1) [37,38,39,40]. It is expected that the combination of naphthalimide and thiourea units will create a novel type of probe which may be utilized for the detection of transition metal ions. In this paper, we explored the possibility of this combination mode and attached thiourea together with carbonyl to 1,8-naphthalimide through an imino group.

Based on this consideration, a new chemosensor *N*-(2-(2-*butyl*-1,3-dioxo-2,3-dihydro-1*H*-benzo[*de*]isoquinolin-6-yl)hydrazine-1-carbonothioyl)benzamide (**L**) was designed and synthesized for the highly selective detection of Co^2+^. The **L** chemosensor detected Co^2+^ by a color change from pink to colorless via the naked eye with high selectivity. In addition, **L**-Co^2+^ showed a remarkable enhancement of the fluorescence intensity in a CH_3_CN/HEPES (4/1, *v*/*v*, pH = 7.4) solution. Moreover, the **L** probe could detect Co^2+^ down to a concentration of 0.26 µM. The qualitative and quantitative analysis of Co^2+^ ions can be realized through color transformation and spectral changes, respectively. It was envisioned that the free **L** emitted weak fluorescence due to the photoinduced electron transfer (PET) and intramolecular charge transfer (ICT) processs from the imino N atom to the naphthalimide units. However, the coordination of Co^2+^ with ligating atoms modulated the electron/charge distribution not only blocked the PET effect, it also reduced the low-energy ICT emission. Last but not the least, the **L** probe was workable and effective for the fluorescence imaging of Co^2+^ in a biological environment. 

## 2. Results and Discussion

### 2.1. UV-Vis and Fluorescence Response of the L Probe Toward Various Metal Ions

The colorimetric selectivity of the **L** sensor was examined first. The absorption response of **L** with various metal ions of Na^+^, K^+^, Ag^+^, Cu^+^, Mg^2+^, Zn^2+^, Ca^2+^, Sn^2+^, Cu^2+^, Ni^2+^, Mn^2+^, Pb^2+^, Cd^2+^, Ba^2+^, Hg^2+^, Fe^2+^, Co^2+^, Fe^3+^, Al^3+^ and Cr^3+^ was carried out in CH_3_CN/HEPES (4/1, *v*/*v*, pH = 7.4). As shown in Figure 1, **L** (1 × 10^−5^ M) itself in CH_3_CN/HEPES (4/1, *v*/*v*, pH = 7.4) showed a pink color and an obvious absorption peak emerged at 510 nm, which may be due to the existence of intramolecular charge transfer (ICT) transition between the nitrogen atom (often an amide group) and the quinone ring [41], which thus caused absorption in the visible region. Upon the addition of five equivalents of Co^2+^, the band at 510 nm decreased sharply, almost disappeared and was accompanied by a weak absorbance under 0.05, which means nearly no absorption was observed in the visible region (420–700 nm). This was because of the coordination of the nitrogen atom and Co^2+^, which resulted in the extremely decreased electron-donating ability of the nitrogen atom involved in coordination; accordingly, this decreased ICT transition. Meanwhile, the absorption band at a short wavelength area (300–420 nm) was redshifted by about 20–50 nm relative to that of **L** and **L** with other cations. This was probably caused by the complexation of Co^2+^, with **L** giving an increased degree of conjugate for the **L**-Co^2+^ complex. The addition of Co^2+^ caused obvious absorbance changes, not only in the UV range but also in the visible range. Through reacting with Co^2+^, **L** exhibited a weak absorption with a color change from pink to colorless which could be easily read out with the naked eye. Other cations had negligible effects on the absorption peak. 

To obtain insight into the fluorescent properties of **L** towards metal ions, these ions’ fluorescence spectra in a CH_3_CN/HEPES (4/1, *v*/*v*, pH = 7.4) solution were investigated. As displayed in Figure 2, **L** (1 × 10^−5^ M) alone had a weak fluorescence emission with broadened photoluminescence, which is consistent with the fluorescence property of intramolecular charge transfer transition [42]. When five equivalents of metal ions such as Na^+^, K^+^, Ag^+^, Cu^+^, Mg^2+^, Zn^2+^, Ca^2+^, Sn^2+^, Cu^2+^, Ni^2+^, Mn^2+^, Pb^2+^, Cd^2+^, Ba^2+^, Hg^2+^, Fe^2+^, Fe^3+^, Al^3+^ and Cr^3+^ was added to the **L** sensor, it was found that the solution of **L** either exhibited no or a small significant increase in fluorescence. By contrast, a 33-fold enhancement of the fluorescence intensity was presented upon the addition of five equivalents of Co^2+^ with a slight blue-shift of fluorescence spectra caused by the suppression of the charge transfer state (Figure 2). The quantum yields of the **L** probe with and without five equivalents of Co^2+^ were 0.63 and 0.02, respectively. when rhodamine B was selected as the standard (*Φ*_F_ = 0.69 in CH_3_CN). Importantly, only Co^2+^ strengthened the fluorescence. The **L** probe nearly emitted no fluorescence in solution (“off” state) as the dark-state of PET and ICT process with inferior radiative transition from the electron-rich amide moiety to the naphthalimide fluorophore occurred [43]. Upon the addition of Co^2+^, the coordination between Co^2+^ and the receptor unit of **L** restrained the PET and ICT processes, directly leading to a strong local-state emission presenting at blue band (“on” state), which is very sensitive to naked eyes (Figure 2 and Figure 3). On the other hand, the low fluorescence of **L** may be due to its flexibility, which contributed to the nonradiative decay in the excited state dynamics. The free rotation of receptor and spacer of the **L** sensor in the presence of Co^2+^ restricted intramolecular motions (RIM) [44], and the cobalt complex became a more rigid, coplanar configuration than the **L** itself, resulting in a chelation enhanced fluorescence (CHEF) effect [45,46] (vide infra, Figure 7). Thus, **L** showed a significant increase of the fluorescence intensity and a distinct color change due to the suppression of PET and ICT processes from the ligand to the chromophore core of naphthalimide, which indicates that the **L** sensor could be used as an efficient fluorescence chemosensor for Co^2+^. These results revealed that **L** compound exhibited the ability to selectively discriminate Co^2+^ from other cations.

### 2.2. Fluorescence and UV-Vis Titration Experiments

To further investigate the chemosensing properties of **L**, a fluorescence titration of **L** with Co^2+^ was performed. When the **L** sensor was titrated with Co^2+^, the fluorescence intensity was significantly increased up to three equivalents of Co^2+^, and then no changes were observed up to five equivalents of Co^2+^ at 450 nm. Changes in the emission intensities of the **L** probe (1 × 10^−5^ mol L^−1^) upon the addition of Co^2+^ (1 × 10^-4^ mol L^−1^) are presented in Figure 3. The fluorescence band centered at 450 nm was progressively increased with the increasing Co^2+^ concentration, giving a linear concentration range of 0–25 µM. 

From the fluorescence titration profiles, the limit of detection (LOD) of **L** for Co^2+^ was calculated to be 0.26 µM based on the equation of 3σ/slope [47]. This is lower than the recommended value for the U.S. (United States Environmental Protection Agency, secondary drinking water regulation) and Australia (National Health and Medical Research Council). Notably, the LOD is quite low as compared to the data reported regarding to other organic chemosensors for sensing of Co^2+^ listed in Table 1 [16,17,18,19,20,21,22,23,24,25]. The main analytical characteristics of the proposed chemosensor for the determination of Co^2+^ were compared with those of some previously reported fluorescent chemosensors, including coumarin, diethylenetriamine, oxadiazole, dinitrobenzene-2,2-dipycolylamine, carboxamide, pyrimidine–pyridine and pyrazine–rhodamine. The proposed **L** fluorescent chemosensor has unique features such as a lower LOD compared with those other detection methods. Therefore, the **L** probe is a promising and potential candidate to detect Co^2+^ ions with high sensitivity.

Subsequently, the UV-Vis titration experiments of the **L** sensor with Co^2+^ ions were studied in CH_3_CN/HEPES (4/1, *v*/*v*, pH = 7.4) to evaluate the sensing sensitivity of the **L** sensor for Co^2+^ ion; the result is shown in Figure 4. In the titration profile, the UV-Vis absorption band at 510 nm appeared and decreased gradually with the increasing of Co^2+^, which resulted from the gradual vanish of intramolecular charge transfer transition mentioned above. A Benesi–Hildebrand plot was drawn using 1/(*A*-*A*_0_) as a function of 1/[Co^2+^] and showed a linear curve (Appendix A). The absorption intensity at 510 nm was linearly related to the Co^2+^ concentration from one to five equivalents of the **L** sensor with a correlation coefficient of *R*^2^ = 0.99305, which indicated that the linear range was very wide and in favor of the detection of Co^2+^ ions. The large correlation coefficient confirmed the 1:1 binding mode between the **L** sensor and Co^2+^, which is consistent with job-plot analysis (vide infra, Appendix A). The associate constant *K*_a_ was calculated from the slope and intercept of the line to be 1.2 × 10^4^ M^−1^, which also indicated that the **L** sensor had a large affinity for Co^2+^ ions.

### 2.3. Anti-Interference Test from Other Metal Ions

Selectivity, which represents the relative fluorescence response for the primary cation over other cations, is one of the most important characteristics of a selective chemosensor. To examine the selectivity for Co^2+^ in a complicated background of potential competing species, the fluorescence enhancement of **L** with Co^2+^ was investigated in the presence of other metal ions (Figure 5). An immediate increase in the emission intensity after the addition of Co^2+^ ions provided further evidence for the high selectivity of **L**. The non-fluorescence property of the product of the **L** probe (10 μM) and five equivalents of Co^2+^ was scarcely influenced by the presence of 15 equivalents of competing ions. The result revealed that **L**, as a potential Co^2+^ fluorescent sensor, could selectively bind with Co^2+^ in the presence of other metal ions.

### 2.4. Effect of pH

pH is a primary factor that affects the probe responses for the purpose of applying to biological and environmental systems. In order to investigate the possibility of using **L** in complex media, the fluorescence spectral response of **L** (10 µM) in the absence and presence of Co^2+^ (five equivalents) in a CH_3_CN/HEPES (4:1, *v*/*v*) solution at different pH conditions was evaluated (Figure 6). The enhancement of fluorescence caused by the addition of Co^2+^ ions was observed in the range of pH 4.0–12.0, thus indicating that the **L** chemosensor could be applied in common pH ranges of environmental analysis, including physiological conditions.

### 2.5. Job’s Plot and Reversibility

In order to determine the binding stoichiometry of the complex, a job-plot analysis was carried out. The difference in fluorescence intensity at 450 nm before (*I*_0_) and after (*I*) the addition of Co^2+^ (*I*_0_-*I*)*X was given as ordinate. The difference reached a maximum when the molar fraction (*x*) of **L** was 0.5 (Appendix A). The Job’s plot analysis of the reaction between **L** and Co^2+^ with a maximum fluorescence intensity revealed that the stoichiometric ratio was 1:1 in the cobalt complex. Furthermore, the MALDI-TOF MS peak at *m*/*z* 519.173 corresponding to [Co(**L**-3H) + H_2_O]^−^ (calculated to be 519.051) further confirms that **L** forms a 1:1 complex with Co^2+^ (Appendix A).

Reversible binding experiments were conducted to justify **L** as a chemosensor (Appendix A). The fluorescence intensities of **L** (1.0 × 10^−5^ mol L^-1^) and Co^2+^ (3.0 × 10^5^ mol L^−1^) at 450 nm decreased or increased with the addition of EDTA (3.0 × 10^−5^ mol L^−1^) and Co^2+^ (3.0 × 10^−5^ mol L^−1^) in sequence. The fluorescence changes were nearly reversible after several cycles with the sequentially alternative addition of Co^2+^ and EDTA (Appendix A). Thus, through treatment with a proper reagent EDTA, the **L** sensor could be simply recyclable toward Co^2+^. Such reversibility and regeneration of the **L** sensor could be applicable for the fabrication of a chemosensor to sense Co^2+^.

### 2.6. Sensing Mechanism

To further study the possible binding mode of **L** and Co^2+^, IR spectroscopy was employed to elucidate the coordination mode. Figure 7 shows the comparison of IR spectra of **L** before and after the addition of Co^2+^. With the introduction of Co^2+^, the sharp peaks at 3391 and 3304 cm^-1^ disappeared. These results suggested that the aromatic amine might involve in the coordination of the **L** complex and Co^2+^, which is consistent with the previous UV analysis. Moreover, the carbonyl groups in the imide moiety may stabilize the complex [48]. As depicted in Scheme 1, it is worth noting that the detecting metal ions of chemosensor BTBN were Fe^3+^ and Pb^2+^, whose proposed sensing mechanism was involved in the nitrogen and sulfur atoms. However, the **L** probe with a similar structure to BTBN (differing merely in one more carbonyl group) detected Co^2+^ with high selectivity and sensitivity. The result indicated the oxygen atom of the carbonyl group of **L** should participate in coordination with Co^2+^.

Since the ^1^H NMR titration of the **L**-Co^2+^ complex could not be measured in instruments, density functional theory (DFT) theoretical calculations of **L** and **L**-Co^2+^ were performed in parallel to experimental study using the Gaussian 09 program to address this issue. Because UV-Vis titration and Job’s plot analyses indicated a 1:1 stoichiometric ratio of **L** reacting with Co^2+^, theoretical calculations were performed under the 1:1 combination. The structures of **L** and **L**-Co^2+^ were optimized using B3LYP and B3LYP/LanL2DZ basis sets, respectively (Figure 8). The energy minimized structure of **L** showed a bent structure with a dihedral angle of 61.5° between thiourea and naphthalimide (Figure 8a). However, the **L**-Co^2+^ complex exhibited a relatively planar structure due to the coordination of Co^2+^, which could change intramolecular torsion to some extent (Figure 8b). The results confirmed the RIM and CHEF effects for **L**-Co^2+^, as mentioned above. Moreover, the frontier molecular orbital analysis reveals that in the **L** HOMO probe is located over the thiourea unit and a large portion of naphthalimide, whereas the LUMO is spread over the naphthalimide unit and a large portion of thiourea, which shows a small degree of orbital separation and provided feasibility of the ICT process (Figure 9a). However, HOMO and LUMO both appear mainly over the coordinate ring and the naphthalimide group with large overlap in **L**-Co^2+^ (Figure 9b). Usually, the wavefunctions of the electron in the HOMO and that in the LUMO overlap significantly, resulting in a large oscillator strength in the normal local excited-state that is suited to producing high-efficiency florescence radiation due to the large orbital overlap [49]; this results in the fluorescence enhancement of **L**-Co^2+^. The calculated HOMO-LUMO energy gap for the **L** probe (3.5810 eV) was more than that of the **L**-Co^2+^ (2.8920 eV), which also indicates a favorable coordination between the **L** probe and Co^2+^. From these observations, it can be further confirmed that ICT and PET could occur between the electron-releasing amino group in thiourea and the electron-withdrawing C=O group in naphthalimide, as stated above, which induces a fluorescence turn off for the **L** probe. The changes in charge density on the N-binding sites of **L** after complexion with Co^2+^ inhibited the PET and ICT process and hence resulted in a fluorescence turn on with slight blue-shift. 

Photoinduced electron transfer (PET) from a donor (D) to an acceptor (A) leads to the formation of a charge-separated state consisting of the donor radical cation and the acceptor radical anion [50,51]. Frontier molecular orbital calculations along with optical experiments on both ON and OFF states are presented in this study (Figure 2, Figure 9 and Figure 10). The molecular structure of the **L** probe is shown in Figure 10a and consists of a D group (thiourea) and an A group (naphthalimide) and meets the general accepted mechanism of an intramolecular PET fluorescent probe (D–A) (Figure 10b) [52,53]. In Figure 10c, the schematic frontier molecular orbital diagrams for PET and its energetic requirements are shown where in the OFF state (up), the HOMO level corresponds to the D-based orbital, so the PET is thermodynamically feasible; in the ON state (down), the HOMO level is the A-based orbital, so the PET is not plausible. Thus, the distribution of HOMO and LUMO for **L** (Figure 9) clarifies the PET-sensing mechanism combined with that shown in Figure 10c. Based on the aforementioned results and the literature [38,54,55], the proposed binding mode is shown in Scheme 2.

### 2.7. Cell Imaging

Since low cell toxicity is a key feature for living cell imaging, the cytotoxicity of **L** was examined by an MTT assay with **L** concentrations from 0 to 30 µM. As shown in Appendix A, even the living cells incubated with 20 μM of **L** for 24 h still kept high survival rate. This result suggest that **L** is suitable for applications in biological systems.

Finally, we further tested the capability of the **L** probe to detect Co^2+^ in HepG2 cells. The HepG2 cells were incubated with **L** (10 μM) for 30 min at 37 °C. Simultaneously, the cells were further raised with different concentrations of Co^2+^ for 30 min and then subjected to images. HepG2 cells treated with **L** were shown in Figure 11. After the addition of 25 μM Co^2+^, a remarkable fluorescence signal could be observed (Figure 11), which was consistent with the titration experiment. Meanwhile, fluorescence imaging figure corresponding bright-field image had a good coincidence degree. Merged images of fluorescence and bright-field images showed that the fluorescence signals were localized in the intracellular area, indicating a subcellular distribution of Co^2+^ and a good cell-membrane permeability of the **L** chemosensor. As shown in Figure 11, this proved that the cells were viable during the experiments. This result indicated that the **L** compound could be effectively used for cell imaging and detected the presence of Co^2+^ in vivo.

## 3. Experimental Section

### 3.1. Materials and Instrumentation

All solvents and reagents were commercially available, and they were used without purification. The FT-IR spectra were recorded on a Bruker ALPHA-T spectrometer (Bruker, Bremen, Germany). The ^1^H NMR and ^13^C NMR spectra were recorded on a Bruker AVANVE 400 MHz spectrometer (Bruker, Bremen, Germany). The high-resolution mass spectrometry (HRMS) was recorded on an FTMS Ultral Apex MS spectrometer (Bruker Daltonics Inc, Colorado Springs, CO, USA). Absorption spectra were collected using a UV-2550 ultraviolet spectrophotometer (Shimadzu, Kyoto, Japan). Fluorescence spectra were obtained using a PerkinElmer LS55 fluorescence spectrometer (PerkinElmer, London, UK). The pH values were measured on a PHS-3C pH meter (Inesa, Beijing, China).

### 3.2. Synthesis of Compound L

As shown in Scheme 3, Compounds 1 and 2 were synthesized following a procedure reported before [56,57]. Compound **L** was synthesized according to a reported method [38] where the final 1,8-naphthalimide derivative could be synthesized via three steps (Scheme 3). A mixture of Compound 2 (0.49 g, 2.2 mmol) and benzoyl isothiocyanate (0.43g, 2.65 mmol) was added into 30 mL acetonitrile in a round-bottom flask and was refluxed for 4 h. The solution was then cooled to room temperature, and the orange precipitate was filtered out and purified by recrystallization from acetonitrile with high yield of 63%. The molecular structure of synthetic compounds was precisely confirmed by FT-IR, ^1^H NMR, ^13^C NMR, and ESI-MS spectroscopies (Appendix A). 

*N*-(2-(2-butyl-1,3-dioxo-2,3-dihydro-1*H*-benzo[*de*]isoquinolin-6-yl)hydrazine-1-carbonothioyl)benzamide (**L**): HRMS (ESI): *m*/*z* calculated for C_24_H_22_N_4_O_3_S (M + H)^+^: 447.1413; found 447.1482; ^1^H-NMR (400 MHz, DMSO-*d*_6_): δ 12.21 (s, Ar–NH, 1H), 11.75 (s, N–NH, 1H), 10.07 (s, C–NH–C, 1H), 8.80–7.01(m, Ar–H, 10H), 4.02 (t, *J* = 7.0Hz, N–CH_2_, 2H), 1.62–1.59 (m, C–CH_2_, 2H), 1.37–1.32 (m, C–CH_2_, 2H), 0.91 (t, *J* = 7.2Hz, C–CH_3_, 3H); ^13^C-NMR (100 MHz, DMSO-*d*_6_): δ: 182.88, 167.57, 164.11, 163.42, 149.46, 133.87, 133.63, 132.510, 132.51, 131.36, 131.36, 129.42, 129.30, 129.02, 128.96, 125.70, 122.56, 119.54, 111.91, 106.16, 40.62, 30.27, 20.32, and 14.25. 

### 3.3. Testing and Calculating Methods

The stock solution of the **L** probe (10^−4^ M) was prepared in CH_3_CN/HEPES (4/1, *v*/*v*, pH = 7.4). The salts used in stock solutions of metal ions with deionized water were KCl, NaCl, MgCl_2_, ZnCl_2_, FeCl_3_, SnCl_2_, CaCl_2_, Al(NO_3_)_3_, CuCl_2_, NiCl_2_, MnCl_2_, CrCl_3_, PbCl_2_, BaCl_2_, CoCl_2_.

During the spectroscopic determination, a stock solution of **L** was diluted with acetonitrile to a final test concentration of 1 × 10^−5^ M. Test solutions for selectivity experiments were prepared by placing 10 mL of the **L** solution (1 × 10^−5^ M) into volumetric flasks followed by adding 50 μL of each metal ion stock solution (0.01 M) with micropipette. The vials were shaken for a few seconds, and then the absorption and fluorescence spectra were recorded. Test solutions for titration experiments were prepared by placing 10 mL of the **L** solution (1 × 10^−5^ M) into volumetric flasks followed by adding an appropriate different molar ratio of the Co^2+^ stock solution (0.01 M) with a micropipette. The vials were shaken for a few seconds, and the absorption and fluorescence spectra were recorded. The excitation wavelength was 380 nm for all measurements of fluorescence spectra, and the excitation and emission slit widths were 10 and 10 nm, respectively. The pH was adjusted to 7.4 through the NaOH solution (1.0 M). In the experiment of pH effect of **L** toward Co^2+^, pH adjustment was made with dilute hydrochloric acid and sodium hydroxide.

### 3.4. Theoretical Calculations Methods

Quantum chemical calculations for ground state optimized structures were conducted by density functional theory (DFT) in a Gaussian 09W program package. The ground state structural elucidation were optimized using DFT-based B3LYP functional where 6-31G(d,p) and LANL2DZ basis sets were used for **L** and a model [**L**-Co(II)] complex, respectively.

### 3.5. Cell Incubation and Cell Imaging

HepG-2 cells were cultured in a 1 × SPP medium (0.003% EDTA ferric sodium salt, 0.1% yeast extract, 0.2% glucose, 1% proteose peptone) at 37 °C. Plated on 6-well plates, the HepG-2 cells were allowed to adhere for 12 h. Afterwards, in order to remove the culture solution, the cells were washed with a phosphate buffer saline (PBS) butter and incubated with the **L** probe (0.5 μM). After pretreatment for 20 min at 37 °C, excess, the **L** probe was gently washed with the PBS butter three times and incubated with Co^2+^ (10 μM) for a further 20 min at an indoor temperature. After washing cells with a prewarmed PBS buffer three times, cell imaging was then carried out. The fluorescence was collected by a Nikon A1 confocal laser microscopy system (Nikon, Japan), and fluorescence images were recorded at green channel.

## 4. Conclusions

In summary, a new chemosensor for Co^2+^ based on 1,8-naphthalimide appending thiourea was successfully designed and synthesized. The **L** probe was essentially insensitive to protons over a wide range of pH (4.0–12.0) while retaining high selectivity and sensitivity toward Co^2+^ in CH_3_CN/HEPES (4/1, *v*/*v*, pH = 7.4). **L** displayed a “turn on” fluorescence emission and a colorimetric response in the presence of Co^2+^, which relied on the binary effects of PET and ICT. It was found that **L** has a low detection limit of 0.26 µM for Co^2+^, which is lower than that of many reported organic chemosensors, and a large number of biologically relevant metal ions exhibited no substantial interference. The UV-Vis titration and Job’s Plot analysis indicated that the complexation stoichiometry between **L** and Co^2+^ was 1:1. Moreover, **L** showed outstanding cell permeation and almost no toxicity upon the selective detection of Co^2+^ in living cells, leading to the possibility of its application as a highly selective probe for biological systems. We expect that the present strategy can provide inspiration for the design of new types of chemosensors for Co^2+^ in biological systems.

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
