# Peer review of "A New Fluorescent Chemosensor for Cobalt(II) Ions in Living Cells Based on 1,8-Naphthalimide"

_molecules, 2019, doi:10.3390/molecules24173093_

Round 1
Reviewer 1 Report
Liu and co-workers submitted the revised manuscript entitled “A new fluorescent chemosensor for cobalt(II) ions in living cells based on 1,8-naphthalimide” to “Molecules (I. F = 3.060)”. In this revised paper, author rectified majority of queries raised earlier. But, still major revision needed before its acceptance.
Solvent selection still not justified clearly. Without NMR titration, binding site confirmation through FTIR data is not fair enough. For 1:1 stoichiometry, solid evidence such as Mass data need to be delivered. The DFT data can be considered as a supportive data not as solid evidence for 1:1 Stoichiometry. Author tend to support the applicability by means of cell image. But, before move on to cell imaging, it is essential to deliver the toxicity data (MTT assay and IC50) of the probe L. In Figure 11, cell images must be delivered in a manner of (a) Bright field (b) Fluorescent and (c) Merged. How many cycles of reversibility attained with EDTA?. Figure S6 should be presented in a readable way.
Reviewer 2 Report
Even though its novelty and relevancy in application to the real systems are somehow questionable, it can be appreciated that many parts of the original manuscript had been improved a lot, particularly application to the live cells and rationalization of fluorescent behavior on cobalt binding. The current form may be acceptible for publication.
Reviewer 3 Report
The manuscript could be accepted after addressing the issues below,
In the job’s plot, a 1:1 ratio was determined. However, in optical measurement, the fluorescence intensity keeps increasing as the addition of Co2+ (0-5 equiv.). Why fluorescence goes up with additional Co2+ in the solution? I have some concerns about the pH buffering ability. Since the measurement condition is CH3CN/HEPES (4/1, v/v), which means that the water fraction is only 20%, how to maintain the buffering ability in the pH effect part? The cytotoxicity of the fluorescent probe should be tested. In the conclusion, the word “hydrosoluble” in “The hydrosoluble probe L was essentially insensitive…” should be changed or deleted since this fluorescent probe is not water soluble. Otherwise, the measurement condition CH3CN/HEPES (4/1, v/v) would not be used.
Round 2
Reviewer 1 Report
Author should rectify the formatting errors and need to improve the resolution of Figures
This manuscript is a resubmission of an earlier submission. The following is a list of the peer review reports and author responses from that submission.
Round 1
Reviewer 1 Report
This manuscript introduces a naphthalimide-based cobalt sensor in a selective manner among various metal ions. However, it has several weak points considering an application to the real system, its novelty, or quality as a paper.
Application to biological or environmental systems would be very hard due to its low solubility in aqueous solutions. In fact, the solvent in this work was 80% acetonitrile. Besides, thiol species or other good chelating species were never been considered in this study.
Regarding its novelty, it is really difficult to agree with the authors’ claim ‘noble sensor’ since many similar works, based on naphthalimides, including their own works have been published for other metal ions.
The quality of writing in this form is not good enough to be published for any journal. Besides, the references ranging 1-8 are not relevant to what is shown in the text.
This work seems to be too preliminary and limited as a publication of a cobalt selective sensor.
Reviewer 2 Report
Y. Liu and co-workers submitted the manuscript entitled “A highly selective fluorescence “turn-on” chemosensor for cobalt(II) ions based on 1,8- naphthalimide with thiourea group” to “Molecules (I. F = 3.060)”. In this paper they presented the naphthalimide based probe for selective detection of Co(II) ions. Though the work seem to be impressive, it have many issues in the current format. Therefore should be rejected in line with the following queries.
1. Why author chosen CH3CN/HEPES (4:1) as a solvent? What happened with other solvents such as ethanol, DMSO etc.
2. Where is the quantum yield values for sensor and probe?
3. At what ratio interference studies were carried out? It should be at 1:3, 1:5 or 1:10.
4. Author does not provide any reversibility data for L + Co(II). Then how come they justified as chemosensor?. Because, it is also possible to fluorescent turn-on through chemodosimeter (reaction based) mechanism
5. Supporting figures should be mentioned in the manuscript, which are not highlighted in the current manuscript.
6. For 1:1 stoichiometry, solid evidence such as Mass data need to be delivered.
7. Binding site should be outlined through 1H NMR titrations along with IR studies presented in Section 2.6.
8. In the current format, authors only delivered the binding approach not mechanism (section 2.6). Author need to clarify sensing mechanism as PET (Scheme 2).
9. The sensing ability of their probe L can be supported by Density Functional Theory (DFT) calculations. Author must deliver HOMO and LUMO of probe L and L + Co(II).
10. Why author omitted important metal ions such as Ag(I), Fe(II), Hg(II), Cd(II)and Pb(II) for interference studies?. It is better to include those ions.
11. Why author did the Stern-Volmer calculations for association constant, which is applicable for fluorescent quenching sensors. For this “Turn-On” sensor, author must calculate binding constant through “Benesi-Hildebrand” equation.
12. Many sections are in short of explanation, which need to be enhanced at their revised version.
13. There is no application part provided for this sensor. Author should support their sensor via cell imaging or real sample analysis.
14. Inset in Figure 2 is not clear, hence should be enhanced with resolution.